Age and growth of Palaeoloxodon huaihoensis from Penghu Channel, Taiwan: significance of their age distribution based on fossils

Kang Jia-Cih 1
Lin Chien-Hsiang chlin.otolith@gmail.com 1 2
Chang Chun-Hsiang chang28@gmail.com cch@nmns.edu.tw 1 3
1 Department of Life Science, Tunghai University , Taichung City , Taiwan
2 Biodiversity Research Center, Academia Sinica , Taipei City , Taiwan
3 Department of Geology, National Museum of Nature Science , Taichung City , Taiwan
Wedel Mathew
Electronic publication date: 2021 Apr 14
Publication date: 2021
Volume: 9
Electronic Location ID: e11236
Received 2021 Jan 7; Accepted 2021 Mar 17
Copyright: ©2021 Kang et al.
Copyright year: 2021
Copyright holder: Kang et al.
License: This is an open access article distributed under the terms of the Creative Commons Attribution License, which permits unrestricted use, distribution, reproduction and adaptation in any medium and for any purpose provided that it is properly attributed. For attribution, the original author(s), title, publication source (PeerJ) and either DOI or URL of the article must be cited.
License URL: https://creativecommons.org/licenses/by/4.0/

Keywords: Age distribution, Pleistocene, Subtropical West Pacific, Elephant age group, Lamellar frequency, Tooth morphology, Taiwan, Penghu Channel

Funding: Taiwan Ministry of Science and Technology MOST 109-2116-M-178-002- 108-2116-M-029-001-MY2 109-2116-M-001-022 This research was financially supported by the Taiwan Ministry of Science and Technology (MOST 109-2116-M-178-002-) (108-2116-M-029-001-MY2, 109-2116-M-001-022). The funders had no role in study design, data collection and analysis, decision to publish, or preparation of the manuscript.

==============================
Dental material attributed to Palaeoloxodon huaihoensis from the Middle to Late Pleistocene were recovered over decades from the Penghu Channel during commercial fisheries activities. The National Museum of Nature Science (NMNS) has a collection of such dental material, which differs in size and morphology and likely represents ontogenetic variation and growth trajectory of various age groups of P. huaihoensis. However, little is known regarding age determination. By using length of dental material, enamel thickness (ET), and plate counts, we established the method to distinguish the age of the species, which is directly derived from the extant African forest elephant Loxodonta africana. When measuring signs of allometric growth, we found that in both the upper and lower jaws, tooth width was correlated negatively with lamellar frequency but positively with ET. In the same age group, the number of lamellae was higher in P. huaihoensis than in L. africana. The reconstructed age distribution indicated no difference in the upper or lower jaw. Notably, within our sample, P. huaihoensis is skewed towards adult and older individuals with median age between 33–34.5 years and differed significantly from that of Mammuthus primigenius in the European Kraków Spadzista site. This age distribution pattern is speculated to be related to the harsh environmental conditions and intense intraspecific competition among P. huaihoensis during the last ice age.

Introduction

The fossil genus Palaeoloxodon (Palaeoloxodontinae, Elephantidae) is widely recorded from Eurasia, Africa, and East Asia during the Late Pleistocene (Makiyama, 1924; Matsumoto, 1929; Osborn, 1936; Zong, 1987; Haynes, 1991). Palaeoloxodon has eight known species: Palaeoloxodon antiquus (Falconer & Cautley, 1847; Osborn, 1942), P. namadicus (Falconer & Cautley, 1847; Osborn, 1924; Matsumoto, 1929), P. falconeri (Falconer, 1862; Busk, 1867; Vaufrey, 1929; Osborn, 1942), P. mnaidriensis (Adams, 1870), P. cypriotes (Bate, 1903; Osborn, 1942), P. recki (Dietrich, 1916; Maglio, 1970; Maglio, 1973), P. naumanni (Makiyama, 1924), and P. huaihoensis (Qi, 1999). In China and neighboring areas, fossil records of Palaeoloxodon are relatively abundant (Liu, 1977; Qi, 1999), and many specimens are assigned to P. namadicus, P. naumanni or P. huaihoensis (Ho, Qi & Chang, 2000; Shieh & Chang, 2007; Qi, 1999). Among the three species, P. namadicus is found mostly in the Nihewan Basin, China (Wei, 1976). Records of P. naumanni are widely distributed in China and Japan but not in Taiwan (Takahashi & Namatsu, 2000). Palaeoloxodon huaihoensis is the only known species from the Penghu Channel, Taiwan (Shieh & Chang, 2007). Ho, Qi & Chang (2000) stated that P. huaihoensis was once distributed both in the China and Taiwan area during the Pleistocene (Shieh & Chang, 2007).

You et al. (1995) divided the Eastern China Sea into three paleobiogeographic zones in the Late Pleistocene, with the north of 38°N representing Mammuthus–Coelodonta fauna, 28°N–38°N representing Palaeoloxodon–Elaphurus davidianus fauna, and Ailuropoda–Stegodon fauna to south of 28°N. According to this scheme, Taiwan and the adjacent Penghu Channel should belong to the Ailuropoda–Stegodon fauna category. However, the Penghu fauna is mainly composed of E. davidianus, Bubalus teilhardi, and P. huaihoensis (Kuo, 1982; Hu & Tao, 1993; Ho, Qi & Chang, 1996; Ho, 1998; Qi, 1999), which is more similar to the fauna in the Huaihe River Region, which belongs to the Palaeoloxodon–E. davidianus fauna (You et al., 1995; Chen, 2000; Ho, Qi & Chang, 2008). Studies indicate that the existence of a narrow and semiclosed sea similar to a land bridge between the Yellow Sea and East Sea in the last ice age (Chen, 2000). Therefore, the paleoclimate in the Pleistocene Taiwan Strait might belong to the tropical-temperate zone (Cai, 1999). Indeed, the so-called “Taiwan Landbridge Fauna” includes at least two distinct faunas during the Middle-Late Pleistocene: one spanning from the Middle to early Late Pleistocene (Chochen fauna) and one confined to the Late Pleistocene (Penghu fauna) (Chen, 2000).

The fauna of Chochen area includes several large mammals, such as Rhinoceros sinensis hayasakai (Hayasakai, 1942), Stegodon (Parastegodon) akashiensis (Hayasakai, 1942; Shikama, Otsuka & Tomida, 1975; Otsuka, 1984), and Stegodon (Parastegodon) aurorae (Shikama, Otsuka & Tomida, 1975), but no fossils of P. huaihoensis were found (Kuo, 1982; Ho & Qi, 1999). The Chochen fauna is believed to share more affinities with that of the Huanan area in southern China than in with the mammal fauna from northern China (Ho, 1998; Cai, 1999; Ho & Qi, 1999; Shieh & Chang, 2007). However, the taphonomic and postmortem transportation processes of Chochen area are very complex and somewhat ambiguous, which resulted in both terrestrial and marine elements in the whole fauna (e.g., Lin et al., 2019). However, the composition of the Penghu fauna indicates that all of it likely originated from northern China throughout the Pleistocene (Ho, Qi & Chang, 1997; Qi, 1999; Shieh & Chang, 2007).

The fossils of elephant teeth provide crucial evidence about the ecosystem in the past. The tooth growth pattern enables inference of the population’s age distribution (Haynes, 1985) and the related habitat distribution across vegetation and climate gradient (Webb, 1977; Janis, 1989; Sukumar, 1992; Fox, 2000; Sukumar, 2003). However, previous works on Palaeoloxodon have reported occurrences only, rather than its age distribution. Therefore, this study explored the age distribution and population structure of P. huaihoensis from Penghu Channel, Taiwan, using the teeth fossils. We categorized the age groups of P. huaihoensis with dental morphological descriptions, reconstructed their age distribution and compared it with other fossil species, and interpreted species distribution in the area.

Materials & Methods

Specimens and measurements

P. huaihoensis specimens were all dredged and recovered by bottom trawling from the Penghu Channel, Taiwan, as in Chang et al. (2015). The Penghu Channel (22°40′N–23°40′N, 119°00′E–120°00′E) is located in the Taiwan Strait between Penghu Island (Pescadores) and Taiwan (Fig. 1). A total of 221 teeth (dp4 (n = 3), M1 (13), M2 (42), and M3 (163)), including 88 mandibles, were available at the National Museum of Nature Science (NMNS), Taiwan for this study. (Fig. 2, Table S1).

Figure 1 Map showing the sampling area in the Penghu Channel (dashed rectangle).

The base map was created using ArcGIS.

Figure 2 Images of P. huaihoensis specimens deposited at the National Museum of Nature Science (NMNS).

(A) Nine enamel loops complete of the lower left dp4 and erosion at both ends, F027933. (B) All lamellae in wear and the lower right M1 is connected to M2, which is slightly worn and lacks enamel thickness (ET), F020284. (C) Nineteen lamellae of the lower left M3 in buccal view, F051590. (D)(E) The upper right and left M3 with all lamellae in wear and slightly eroded at both ends, F026947. (F) Buccal surface of the lower right M3, F020284. (G) Anterior 2–3 enamel loops confluent on the occlusal surface of lower right M3 from catalog number F020226. (H) Lingual view of the lower right M3, F020248. All scale bars represent 5 cm.

We first used the plate counts to identify the dental position of the molar. Next, the tooth length, width, and height were measured (Fig. 3), with the height taken vertically from the crown apex of the plate. The enamel thickness (ET) was measured with calipers. To calculate lamellar frequency, the number of complete plates at 10 cm at the crown base of both the lingual and buccal sides was taken (Short, 1969; Hasegawa, 1972; Maglio, 1973; Shieh & Chang, 2007).

Figure 3 Measurements of an elephant tooth used in this study.

(A) The length of dental material and width were measured. (B) The height of tooth was taken from the crown apex of the highest plate to the crown base on both the lingual and buccal sides.

Age determination

We used the size, wear of teeth and dental morphology to determine the age distribution of P. huaihoensis (Morrison-scott, 1947; Sikes, 1968; Maglio, 1973; Lang, 1980). Thirty age groups based on tooth morphology and shearing rate of deciduous teeth of African forest elephants were established by Laws (1966), and this method has been widely used for the reconstruction of age distribution in many elephant species (Haynes, 1991; Lister, 1999). We used this method too with slight modifications. For example, Laws’ method indicates that M3 has a maximum number of 12 plates in L. africana, but in P. huaihoensis, as many as 22 plates can be found in M3. In this case, the remaining number of plates in P. huaihoensis can be obtained by the rate of tooth eruption of L. africana multiplied by the observed plates of P. huaihoensis. Thus, the age group XX of Law’s with 12 plates indicates that there will be six plates in the age groups of P. huaihoensis if (22/12) ×6 = 11 plates are remaining (see Table S2). The rate of tooth eruption of L. africana represent the value of ratio that expect the appeared number of plates of P. huaihoensis. Consequently, we established 24 age groups defined using 88 jaws (Fig. 4).

Figure 4 Definition of age groups I–XXIV.

I, dp4 all lamellae in wear, M1 slight wear (specimen number: F027933); II, dp4 well worn, approximately 3-4 plates remaining; M1 first 1-2 lamellae in wear (F051613); III, M1 all in wear; M2 worn to enamel of first two lamellae (F044264); IV, M1 first 1-2 enamel loops confluent, M2 slight wear (F020284); V, M1 well worn; M2 more enamel loops showing (F051497); VI, M1 only 5-6 enamel loops left, slight erosion of posterior border; M2 lamellae well formed (F051562); VII, M1 well worn, only three plates remain; M2 slight erosion of anterior edge, 9-10 enamel loops complete (F027950); VIII, M2 first enamel loops confluent (F044271); IX, M1 worn out; M2 well into wear showing lozenges, more lamellae visible (F020247); X, M2 all except last 3 lamellae in wear (F020255); XI, M2 complete, all lamellae in wear, and all enamel loops showing M2 erosion at both ends; M3 lamellae well formed (F027988); XII, M2 all lamellae in wear, 15 enamel loops complete (F026927); XIII, M2 only approximately 8-9 loops remain and erosion at both ends (F020287); XIV, M3 worn to enamel of first lamellae and more enamel loops (F030111); XV, M2 lost; M3 11-12 enamel loops complete (F020278); XVI, M2 worn out; M3 no erosion of anterior border, anterior 1-2 enamel loops confluent (F044257); XVII, M3 only 2 lamellae not in wear (F027320); XVIII, M3 all except last lamellae in wear (F044266); XIX, M3 first 1-2 enamel loops may confluent (F051487); XX-I, M3 erosion at both borders, anterior 2-3 enamel loops confluent (F026942); XX-II, M3 all except last lamellae in wear (F020258); XXI-I, M3 more enamel loops showing, slight erosion of the anterior border (F044270); XXI-II, M3 well worn, first enamel loops may be slightly confluent (F051560); XXII-I, M3 all lamellae in wear, no erosion at both ends (F044268); XXII-II, M3 erosion at both borders, anterior 2-3 enamel loops confluent (F027963); XXIII-I, M3 only five complete enamel loops remain, anterior part broken off (F044261); XXIII-II, anterior third of tooth missing, only five complete lamellae remain (F027967); XXIV, M3 only 2-3 loops remain (F051559).

Statistical analysis

The tooth width and lamellar frequency in occlusal and buccal sides of the lower and upper jaws of dp4-M3 as well as the relationship between the width and enamel thickness (ET) of lower and upper jaws of dp4-M3 were plotted using R software (R Core Team, 2013). The relationship between two variables was indicated using Pearson’s correlation coefficient. These relationships reflect whether the concerned variables revealed an allometric growth pattern. The number of lamellae throughout the lifespan was plotted against the estimated age of P. huaihoensis (see above, Age determination), and these were directly compared with those of L. africana (Laws, 1966).

A histogram based on the frequency distribution of specimens was established to reconstruct the age distribution of P. huaihoensis. Unlike studies in which only the lower mandibles were considered (Laws, 1966), we included upper mandibles specimens for comparison. A null hypothesis of the distributions of upper and lower jaws was first tested using the two-sample t test. However, when no significant difference between upper and lower mandibles was detected, only lower jaw specimens were used in subsequent analyses. A Shapiro–Wilk test was conducted to test whether the fossil age distribution data were distributed normally; if not, the median for the lower jaws was calculated using the Wilcoxon–Mann–Whitney test.

Finally, we compared the age distribution based on fossil remains of P. huaihoensis with other species: the stable age distribution of fossil Mammuthus primigenius and Mammuthus columbi. A null hypothesis stating the same age distribution for each population pair was analyzed using Pearson’s chi-square test. Here, the independence of age and the number of individuals in each of the two populations were tested. The M. columbi and M. primigenius data were derived from the studies of Louguet-Lefebvre (2013) and Wojtal (2001), respectively. All analyses were performed using R (Core Team and Others, 2013).

Results

Tooth width and lamellar frequency were negatively correlated on both the occlusal and buccal sides for dp4-M3. Lamellar frequency increased when tooth width decreased in both upper and lower jaws (Figs. 5A, 5B, 5D and 5E). By contrast, the tooth width and ET were positively correlated on both the sides (Figs. 5C and 5F). The size range overlapped in some cases; for instance, the M2 overlapped with M3 in occlusal width and lamellar frequency and width and ET of the lower jaw, respectively (Figs. 5D and 5F).

Figure 5 The relationships of various meristic measurements in the jaws of dp4-M3.

(A) Tooth width and lamellar frequency in the occlusal surface of the upper jaw (r =  − 0.558, t =  − 7.699, p < 0.05). (B) Tooth width and lamellar frequency in the buccal side of the upper jaw (r =  − 0.476, t =  − 6.201, p < 0.05). (C) Tooth width and enamel thickness (ET) of the upper jaw (r = 0.531, t = 7.179, p < 0.05). (D) Width and lamellar frequency in the occlusal surface of the lower jaw (r =  − 0.649, t =  − 7.915, p < 0.05). (E) Width and lamellar frequency in the buccal side of the lower jaw (r =  − 0.453, t =  − 7.523, p < 0.05). (F) Width and ET of the lower jaw (r = 0.457, t = 4.759, p < 0.05).

A summary of the various age groups derived from the tooth morphology, lamellar number, teeth position, and age estimation is presented in Table 1. The number of lamellae of P. huaihoensis was considerably higher than that of L. africana in the same age group (Fig. 6). Moreover, the increasing rate of lamellae in P. huaihoensis was progressively more evident than that of L. africana from M1, eventually reaching 22 lamellae in M3.

Table 1 Comparison of estimated ages derived from the lower jaw of P. huaihoensis and L. africana. The positions of the teeth used in Laws (1966) are indicated in parentheses.

Tooth position	L. africana (from Laws, 1966)	P. huaihoensis (this study)	
	Age groups	No. of lamellae	Age (yrs)	Age groups	No. of lamellae	Age (yrs)	
dp2	I–V	3	0–3	–	–	–	
dp3	VI–X	7	4–13	–	–	–	
dp4	XI–XV	9	15–24	I–IV	9	4–16	
M1	XVI–XX	9	26–34	V–X	11	18–28	
M2	XXI–XXV	10	36–47	XI–XVI	17	32–41	
M3	XXVI–XXX	12	49–60	XVII–XXIV	22	43–57	

Figure 6 Differences in the relationship of the number of lamellae and age in P. huaihoensis and L. africana. Data of L. africana are from Laws (1966).

The reconstructed age distribution of P. huaihoensis revealed that the age peaked at 29–36 years, indicating a higher number of adult individuals (Fig. 7). Notably, the distributions of the upper and lower jaws were similar (two-sample t test, p = 0.941, t = 0.075), and they possibly originated from a single population (mean = 0.04). The Shapiro–Wilk test indicated a nonnormal age distribution (p < 0.05), and using the Wilcoxon–Mann–Whitney test, the medians of lower jaws indicated an age of 33–34.5 years.

Figure 7 Age distribution of P. huaihoensis from Penghu Channel, Taiwan. The frequency (%) is based on the proportion of specimens (n).

Pearson’s chi-square test revealed that P. huaihoensis age distribution was significantly different from the stable age distribution of M. primigenius (p < 0.05, Fig. 8A) but not from that of M. columbi (p > 0.05, Fig. 8B). M. primigenius mainly comprised juveniles and young-adult individuals, whereas P. huaihoensis and M. columbi comprised mostly adults aged 30–40 years.

Figure 8 Comparison of the age distribution of P. huaihoensis with that of (A) M. primigenius and (B) M. columbi.

Discussion

Tooth eruption has widely been used for estimating extant elephant age (Laws, 1966; Krumery & Buss, 1968; Shoshani, 2002; Roth & Shoshani, 1988). This method has also been applied to fossil species—for example, the age distribution of the Mammut (Mastodon) (Haynes, 1985), M. columbi (Saunders, 1980; Louguet-Lefebvre, 2013), and M. primigenius (Lister, 1999; Wojtal, 2001; Rountrey et al., 2012). However, in P. huaihoensis, plate count, length, ET, and lamellar frequency measurements revealed substantial differences from the extant L. africana (e.g., Fig. 6).

Our age distribution for P. huaihoensis has a distinct pattern compared with that of M. primigenius (Wojtal, 2001). In M. primigenius, numerous younger individuals (0–12 years) and fewer adults were found in the European Kraków Spadzista site (Fig. 8A). The pattern of M. primigenius represents the natural deaths of the whole population, suggesting nonselective cumulative deaths in the normal environment (Klein, 1985; Haynes, 1991; Haynes & Klimowicz, 2016).

Although the upper and lower jaws of P. huaihoensis suggest that these specimens originate from a single population, the reconstructed age distribution indicates an older adult–dominant pattern (median = 33–34.5 years). The age profile of P. huaihoensis seems to be similar to that of M. columbi (Fig. 8B), but the living environment and taphonomic process for both species were completely disparate. The Hot Springs site has yielded many specimens of M. columbi, and this site was not only essential for providing a water source for animals inhabiting adjacent areas but also a natural trap with unstable sediments that preferentially traps larger adult individuals (Agenbroad & Mead, 1994). This may be the reason that the inferred M. columbi population mainly comprised adult individuals (Louguet-Lefebvre, 2013). Intense intraspecific competition between adults under harsh environmental conditions can cause massive death; we speculate that this was one of the possible cases of P. huaihoensis. During the last ice age, climate change-related resource shortages likely resulted in sharp competition within the population of P. huaihoensis, particularly in large adult males (Valeix, Chamaillé-Jammes & Fritz, 2007; Ferry et al., 2016).

In addition to competition, the notable older age predominance may have been caused by sampling bias because our materials were collected by bottom trawl fisheries and smaller teeth of P. huaihoensis from younger individuals may not have been sufficiently represented. However, fossils from the Penghu Channel have been collected for decades and have resulted in a massive collection of a diverse fauna (e.g., Hu & Tao, 1993), including fossil remains of much smaller sizes such as fragments of the tibia, vertebrae, ribs, and even a tiny lower jaw of Homo (Chang et al., 2015) were recovered using this method. In any case, small teeth of P. huaihoensis would be considerably represented if they existed. Therefore, the age frequency distribution suggests that the area around Penghu Channel might not have been a nursery ground for P. huaihoensis. Nevertheless, whether our material represents an equilibrium age distribution of P. huaihoensis remains uncertain because this age distribution could have existed only in fossil species.

The fossil records of P. huaihoensis date from the Middle to Late Pleistocene (Liu, 1977; Chen, 2000). The species was first found in the northern part of Anhui, China (Liu, 1977). The further geographical distribution includes Huaihe River Region (Cai, 1999; Ho & Qi, 1999) and northern Jiangsu, China (Qian et al., 2017; Chen et al., 2020) (Fig. 9). In Taiwan, however, the species has only been found in the Penghu Channel and never southwards; thus, it is not found in the famous Chochen fauna (Kuo, 1982). Because of cold temperatures and water and food shortage, animals could have migrated from higher to lower latitudes; in particular, P. huaihoensis could have migrated southward in search of grasslands and water resources (Webb, 1977; Janis, 1989; Fox, 2000). However, possible ecological explanations, such as climate change and niche competition, have yet to be explored fully. On the other hand, this age distribution—adult-dominated, and with young individuals being rare—is not that uncommon among proboscideans (see Green & Hulbert Jr, 2005 on mastodons), and it may well be possible that there are a number of causes that could independently lead to this age structure. For example, the differences between the age-related populations of M. columbi at Hot Springs, M. primigenius in the European Kraków Spadzista site, and P. huaihoensis from Penghu Channel, could also due to unrecognizable time averaging effect. Overall, the fossil records suggest that P. huaihoensis was distributed from northern China and to as far south as Penghu Channel in the last ice age but did not migrate across the Taiwan Strait to Taiwan Island (Fig. 9).

Figure 9 Postulated migration direction (black arrow) of P. huaihoensis.

The species likely originated from northern China (white circle), where fossil records are more abundant. The extension of the record in the Penghu Channel (black rectangle) in the last ice age is currently its southern limit. The current sea depth contour (−120 m) delineates the ancient coastline during the last ice age. The map is derived from the National Centers for Environmental Information (https://www.ngdc.noaa.gov).

Conclusions

The age distribution of such a large mammal as P. huaihoensis, which once inhabited the subtropical West Pacific in the Late Pleistocene, has been largely unknown. By using its dental material from the Penghu Channel, we reconstructed its age distribution and defined 24 age groups by measuring the ontogenetic morphological changes in teeth length, ET, and plate counts. Compared with M. primigenius, P. huaihoensis from the Penghu Channel is distinct in having significantly more adult and older adult individuals and very few juveniles, similar instead to M. columbi. However, unlike taphonomic patterns of age distribution observed in the case of M. columbi, we speculate that environmental conditions and intraspecific competition are several of the possible causes. The fossil records further indicate that P. huaihoensis was mainly distributed in northern China and only extended southward in the Penghu Channel. The postulated ancient migration route of the species and the possible underlying ecological reasons would benefit from further investigation of the collection from northern China. Future studies should elucidate the exact age distribution of P. huaihoensis in northern China compared with that of the Penghu Channel and conduct isotope analyses to explore the possible vegetation and climatic impacts on the migration and specific age distribution recovered from the Penghu Channel.

Supplemental Information

Supplemental Information 1 Dental measurements of the Late Pleistocene Palaeoloxodon huaihoensis from Penghu Channel

Click here for additional data file.

Supplemental Information 2 Age determination of Palaeoloxodon huaihoensis.

Click here for additional data file.

We are grateful to Prof. Yi-Ching Lin (Department of Life Science, Tunghai University, Taiwan) and Xinyue Ou (Tunghai University, Taiwan) for their constructive comments and suggestions on the statistical analyses.

Additional Information and Declarations

Competing Interests

Author Contributions

Data Availability

The authors declare there are no competing interests.

Jia-Cih Kang, Chien-Hsiang Lin and Chun-Hsiang Chang conceived and designed the experiments, performed the experiments, analyzed the data, prepared figures and/or tables, authored or reviewed drafts of the paper, and approved the final draft.

The following information was supplied regarding data availability:

Dental measurements of the Late Pleistocene Palaeoloxodon huaihoensis from Penghu Channel and the age determination of Palaeoloxodon huaihoensis are available in the Supplemental Tables.

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
