# Peer review of "Age and growth of Palaeoloxodon huaihoensis from Penghu Channel, Taiwan: significance of their age distribution based on fossils"

_PeerJ, doi:10.7717/peerj.11236_

## Round 0.1 · original submission · Minor Revisions

Congratulations, both reviewers found your work to be generally solid and worthy of publication. Both recommended that your manuscript be accepted pending minor revisions, and I agree. I've carefully read the suggestions of both reviewers and their comments strike me as apt and constructive. Please be diligent in addressing the concerns that they have raised. I look forward to seeing an improved version of this work soon.

·

Basic reporting

This paper describes a large sample of teeth from Palaeoloxodon huaihoensis recovered from the Penghu Channel. The paper is well-written and researched. Included figures are clear and appropriate. Some minor points are listed below:

I found the tooth notation to be a bit unusual. Older literature tended to use superscript and subscript numbers to designate upper and lower teeth, respectively. Modern literature generally uses uppercase and lowercase letters (M3 vs. m3, dP4 vs. dp4), and I would suggest using this notation. Related, the measurement supplemental table uses another scheme that is not explained, but I think uses M3 vs. 3M. Again, I suggest uppercase vs. lowercase.

On Figure 3, I suggest adding a diagram showing how height was measured.

In Line 256, “Hot spring” should be “Hot Springs”.

Experimental design

The authors’ methods are observational, and well-described, including basic measurements, age determinations, and basic statistical analysis of the sample. Specific comments are below:

The authors develop a tooth aging scheme for Palaeoloxodon based on the methods of Laws 1966. However, because their sample does not include any young individuals, their Group I starts with 6 year olds. However, presumably younger Palaeoloxodon are known from somewhere. If the authors want to have their scheme widely adopted by other Palaeoloxodon workers, they may want to consider recalibrating their scheme to start at something like VI instead of I, to allow the younger ages to be designated at some future date.

Related to this, the age determination supplemental file attempts, in part, to show how the Palaeoloxodon groups compare to Laws groups. Unfortunately, I found this table very confusing and was not able to completely decipher it. I think this table needs a detailed description explaining it, but I was unable to find one.

Validity of the findings

The bulk of the discussion attempts to explain the sample’s predominance of older individuals in the sample, and the complete absence of very young individuals. The authors make a good case that this is not a sampling bias, and point out that this pattern is similar to that seen in mammoths at Hot Springs, even though the taphonomic settings are completely different.

From these observations, the authors suggest that the age distribution is due to environmental stress leading to intraspecific competition (they say “interspecific” in Line 260, but I think they meant “intraspecific”.) While this is a perfectly good hypothesis, I think it’s a bit of a reach. My impression is that this age distribution (rare young, middle-age and elderly dominated) is not that uncommon among proboscideans (see, for example, Saunders 1977 and Green and Hulbert 2005 on mastodons). It may well be possible (even probable) that there are a number of settings that could independently lead to this population structure. I suggest that the authors soften this conclusion and emphasize that they’re putting this forward as one hypothesis to explain this distribution.

Additional comments

The authors should be commended for publishing these rather obscure but important specimens, and making their data open access so that they can contribute to other studies.

·

Basic reporting

Overall this is a well-written article with a robust set of appropriate statistical analyses to address the proposed hypotheses. There are a few places, especially in the introduction, were there is some awkward wording and passive voice. I would like to see a better discussion of the Hot Spring comparative site. While I agree this is an appropriate assemblage to use in comparison, those not familiar with Hot Springs would be unable to infer some comparisons as is. Will the author's photographs be available or do museum collection permissions make that impossible? While the manuscript is sufficient as is, if possible to have images in supplementary material I would find that ideal. All analyses were well laid out and appropriate for testing the author's hypotheses. The author's examination of possible sources of bias was well done and presents a compelling argument.

Experimental design

comments concerning this were addressed in the previous section. Overall appropriate and well-done statistical analyses, that most certainly address interesting and unique study questions. This clear hypothesis to test, to discussion of results leaves me eager to see potential follow ups to this work.

Validity of the findings

Appropriate statistical analyses were used throughout. Clear explanations of both the reasoning behind selection of particular statistical methods and metrics, as well as well examined sources of possible bias.

Additional comments

While the content of your figures is appropriate and necessary in each case, many are somewhat difficult to read. Larger text, clearer symbols and colors, and additional in-figure labels and expanded figure captions would greatly improve the overall polish of this work.

---

## Round 0.2 · accepted · Accept

Thank you for your diligence in addressing the concerns of the reviewers. I am satisfied with the revised manuscript, and I am happy to accept it for publication in PeerJ.

In general the English is good, but there are a few awkward spots still. For example, the sentence, "On the other hand, this age distribution—that rare young but adult dominated—is not that uncommon among proboscideans" would read better like so: "On the other hand, this age distribution--adult-dominated, and with young individuals being rare--is not that uncommon among proboscideans". One trick that I recommend to most authors is to read their manuscript out loud to catch any awkward sentences. You might consider doing that before you upload the final version of the manuscript for publication.

The decision of whether or not to publish the peer reviews alongside the paper is entirely yours, and will not affect how your paper is handled going forward. However, I encourage you to do so. Making the reviews public allows the reviewers to receive credit for their efforts, and also contributes to the emerging culture of fairness and transparency in editing and peer review.